**Data Availability Statement:** All research files are available at https://doi.org/10.5281/zenodo.10253131

# "An app is just available at all times"—the process and outcomes of converting the *Georgia Tuberculosis Reference Guide* into a mobile application

Santiago J. Arconada Alvarez[1,2], Alison T. Hoover[1,3], Morgan Greenleaf[1,2], Susan M. Ray[1,3,4,5], Marcos C. Schechter[1,3,4,5], Henry M. Blumberg[1,3,4,6], Wilbur A. Lam[1,2,7,8,9]*

1 Emory University School of Medicine, Atlanta, Georgia, United States of America, 2 AppHatchery, Georgia Clinical and Translational Science Alliance, Atlanta, Georgia, United States of America, 3 Department of Medicine, Division of Infectious Diseases, Emory University School of Medicine, Atlanta, Georgia, United States of America, 4 Grady Memorial Hospital, Atlanta, Georgia, United States of America, 5 Georgia Department of Public Health Tuberculosis Program, Atlanta, Georgia, United States of America, 6 Departments of Epidemiology and Global Health, Emory University Rollins School of Public Health, Atlanta, Georgia, United States of America, 7 Department of Pediatrics, Emory University School of Medicine, Atlanta, Georgia, United States of America, 8 Wallace H. Coulter Department of Biomedical Engineering, Emory University and Georgia Institute of Technology, Atlanta, Georgia, United States of America, 9 Aflac Cancer & Blood Disorders Center, Children's Healthcare of Atlanta, Atlanta, Georgia, United States of America

* wilbur.lam@emory.edu

## Abstract

### Background

The physical, paper-based *Georgia TB Reference Guide* has served as the clinical reference handbook on tuberculosis (TB) diagnostic and treatment guidelines for the state of Georgia in the United States. Supported by the Georgia Department of Public Health, the production of the 112-page palm-sized booklet was previously led by a team of Georgia-based TB experts at Emory University and printed every three-five years with updates to clinical management guidelines and TB consult contact information. However, the costs associated with editorial printing combined with delays in updating a static printed booklet with revised guidance hampered the utility of the tool. Considering the barriers with paper-based production and based on the beneficial use of apps to support the dissemination of clinical management guidance in other settings, the booklet was converted into a mobile application. This paper describes the process of developing a mobile app version of the *Georgia TB Reference Guide* in an easy-to-update and readily available format.

### Methods

We employed a user-centered design approach to develop the app, including a series of qualitative interviews and quantitative surveys. Participants included a mix of state officials and local TB experts. First, initial foundational interviews were conducted to conceptualize current utilization practices of both the paper and PDF versions of the tool. Second, the

**Funding:** HB was awarded Contract 40500-046-21203197 from the Georgia Department of Public Health through Contract 40500-046-21203197 which also supports AH. https://dph.georgia.gov/health-topics/tuberculosis-tb-prevention-and-control/tb-publications-reports-manuals-and-guidelines Authors SJAA, MG, HB, and WL are supported by the National Center for Advancing Translational Sciences of the National Institutes of Health under Award Number UL1TR002378. The content is solely the responsibility of the authors and does not necessarily represent the official views of the National Institutes of Health. The funders had no role in study design, data collection and analysis, decision to publish, or preparation of the manuscript.

**Competing interests:** The authors have declared that no competing interests exist.

findings from the initial interviews were organized thematically and informed the design of the app, which was then beta tested by a round of previously unsampled TB experts as well as a re-sample from the initial interviews. Third, the designs were coded into developmental phases and beta tested among users of the current *Georgia TB Reference Guide*. Fourth, the app was published and downloaded by a pre-selected group of local users who provided answers to a follow-up survey after using the app for one month. Fifth, user growth, self-reported demographics, and app usage between February and July 2022 were recorded through automatic data metrics built into the app.

## Results

The paper copy *Georgia TB Reference Guide* usage themes included commonly referenced content, navigation paths, and desired features and content. The themes were converted into features and designs such as prioritizing commonly reviewed topics and guide customization with bookmarks and notes. Iterations of the designs were driven by feedback from TB experts and included home page featured content, improving content readability, and improving the search feature. The follow-up survey revealed a 90% preference for the app over the paper version of the guide. In the six months following the app's release, the app was downloaded by 281 individuals in the United States. The majority of downloads were in Georgia and the app also expanded organically to 19 other states.

## Conclusion

The experience of converting the *Georgia TB Reference Guide* offers specific and effective steps to converting a medical reference guide into a mobile application tool that is readily available, easy to use, and easy to update. The organic dissemination of the app beyond the state of Georgia's borders within the first six months of app launch underscores desire among TB healthcare professionals for high-quality digital reference content outside the state. This experience offers clear outlines for replication in other contexts and demonstrates the utility of similar mobile medical reference tools.

## Introduction

Despite being one of the oldest known diseases in humans, tuberculosis (TB) can be a complicated disease to diagnose and treat [1]. The World Health Organization has estimated that were 10.6 million new TB cases in 2022 and 1.3 million deaths due to TB [2]. In 2023, TB re-emerged as the leading cause of death due to an infectious disease exceeding HIV and COVID-19. The vast majority of TB cases occur in high burden low- and middle-income countries. However, the US Centers for Disease Control and Prevention (CDC) reported 8,331 TB cases in the US in 2022 [3]. TB is caused by *Mycobacterium tuberculosis* and transmitted person-to-person via an airborne route. Infection with *M. tuberculosis* results in a spectrum of outcomes including latent TB infection (LTBI) and active TB disease. Treatment of LTBI among high risk individuals can prevent the progression to active disease [4]. Prompt diagnosis of active TB disease is essential to ensure appropriate therapy is initiated and to prevent severe morbidity and mortality.

There have been several advances in TB diagnostics in recent years, including rapid molecular tests for TB diagnosis and sequencing technology to identify drug resistance, though the reference diagnostic standard remains culture-based methods that take weeks to provide results [5]. There have also been advances in TB treatment, especially in the area of the treatment of drug-resistant TB, with the use of new and repurposed drugs as part of all oral regimens that have led to high cure rates with 6 to 9 months of treatment compared to previous 18–24 month regimens which included an injectable drug [6]. There has also been the development of a 4-month treatment regimen for drug susceptible tuberculosis as well as the development of shorter regimens (1 month, 3 months including a weekly regimen, and 4 months) for the treatment of LTBI [7]. Correctly diagnosing TB, determining drug susceptibility or resistance, and timely initiation of appropriate treatment is critical to reducing mortality and progressing toward TB elimination targets. National and international guidelines have been published on TB diagnosis and treatment, but these may not take local epidemiology and local TB program resources into account [8–11]. The various guidelines can make it difficult for healthcare providers to stay current on recommended diagnostic and treatment protocols for patients with suspected or proven TB disease and there may be delays between the time of important advances in diagnosis and treatment and published guidelines.

In the state of Georgia in the United States, the Georgia Department of Public Health (GDPH) has provided guidance on TB treatment and diagnosis via the *Georgia TB Reference Guide*, which was developed in collaboration with Emory University and based on national guidelines issued by CDC. The *Guide* was originally a palm-sized booklet that fit into the pocket of a clinician's white coat and included topics covering epidemiology of TB in Georgia, diagnosis and treatment of LTBI and active TB disease, infection control guidelines, and reporting guidelines for TB cases in the state of Georgia, as well as local health department contact information [12]. The *Guide* was updated every few years as national recommendations were updated; 8,000–10,000 printed copies were disseminated throughout the state for each new edition. The *Guide* was also uploaded as a portable document format (PDF) file and could be downloaded from the GDPH website. Comprehensive dissemination was challenging, especially for rural areas, and the costs of printing and distribution prohibited small and frequent updates.

Mobile applications have increasingly become a tool for clinician reference guides given their ease in dissemination and rapid ability to update content as guidelines evolve. Mobile applications have shown benefit in increasing confidence and educating physicians in various diagnostic and treatment settings [13–16]. A survey of the available mobile apps related to tuberculosis found that although the number of available apps increased, most of them contained errors, or provided wrong or harmful information [17]. This review identified incorrect descriptions of tuberculosis describing it as diabetes mellitus, erroneous medical advice such as alluding to apple custard remedies, lacking up-to-date information, and spelling or grammatical mistakes. This identified the existence of a need for high-quality clinician-driven information.

To address this need and provide access to the *Guide* by clinicians in a more user-friendly format which could more easily be updated, we focused on developing a mobile application version of the *Georgia TB Reference Guide*. A 'user-centered' approach underpinned our methodology, emphasizing the importance of understanding and incorporating the needs, preferences, and experiences of the end-users; in this case, healthcare professionals providing care to persons with active TB disease and LTBI. This approach is characterized by active and continuous engagement with users through qualitative interviews and quantitative surveys. This engagement is directed at understanding current users as well as designing, developing, and validating the utility of the mobile application [18].

In this paper we describe our process, as well as the key findings from our user-centered approach. We believe our methods can serve as a model for the development of mobile reference material in other disease areas and contribute to the growing body of literature on the use of mobile technologies to improve healthcare delivery and outcomes. Ultimately, our hope is that the *Georgia TB Reference Guide* mobile app and its development process can be widely adopted and help to improve the quality of TB care in Georgia and the United States.

## Methods

### Study design

The *Georgia TB Reference Guide* app was designed and developed following a user-centered design framework [19]. The project was divided into five phases (Fig 1): 1) interviews with paper *Guide* users to conceptualize utilization practices, 2) design evaluation interviews after thematically organizing interview findings and creating a preliminary app design, 3) two rounds of beta testing surveys with an early version of the app, 4) survey of paper-based guide users once the app had been publicly launched in application stores, and 5) in-app survey to collect self-reported user demographics, and broad collection of usage data in the app such as most-used content and geographic distribution.

### Participants

Phases 1 through 3 of the project involved a total of ten TB coordinators at public health clinics across nine health districts in Georgia, four infectious disease (ID) fellows at Emory University, and one attending physician (Infectious Diseases) at Emory. Phase 4 involved 30 Georgia healthcare workers including some who were sampled in phases 1 through 3. Participants were purposively sampled via identification from authors HB and SR and recruited via email. Phase 5 involved a total of 281 participants, 61 of which self-reported demographic information. All participants consented verbally to participating in the research. The project was reviewed by the Emory University Institutional Review Board (IRB) and determined to not require IRB review because it did not meet the definition of "research" with human subjects or "clinical investigation".

### Phase 1: Interviews

A group of pre-selected individuals were recruited via email to participate in the first round of interviews (n = 9, 7 TB coordinators working in public health and 2 Emory Infectious Diseases fellows). Interviews were conducted remotely using Zoom (San Jose, CA) and lasted approximately one hour. Participants described their experiences with the paper version of the *Georgia TB Reference Guide* related to managing cases of LTBI and active TB disease. The interviews followed a semi-structured format and covered eight topics outlined in Table 1.

The interview notes were compiled and organized using Miro (Miro Inc, San Francisco, CA), a digital white canvas software tool. Common insights from the interviews were grouped together spatially and then assigned themes following the affinity diagram technique [20] and inductive analysis [21]. Themes were translated into design implications and low-detail designs were produced using the digital design tool Figma (Figma Inc, San Francisco CA). Low-detail designs, often referred to as low-fidelity designs, are preliminary sketches that focus on the basic layout and flow of the application. These designs prioritize the spatial arrangement and functionality of key interface elements without delving into detailed graphics or aesthetics.

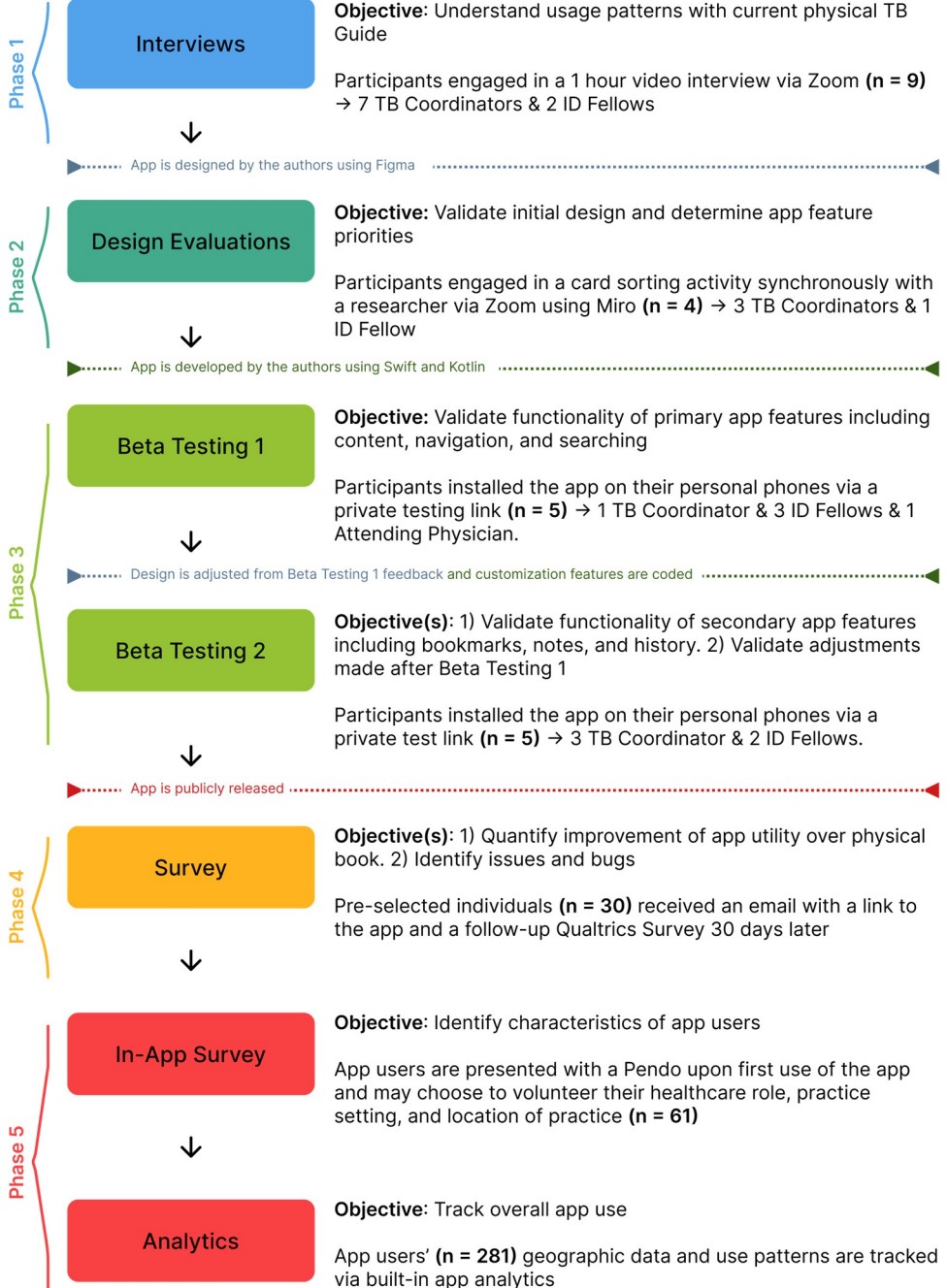

**Fig 1. Flow chart of the project's methodology.** Miro (Miro Inc, San Francisco, CA). Figma (Figma Inc, San Francisco CA). Kotlin (Google Inc, Palo Alto CA). Swift (Apple Inc, Cupertino CA).

## Phase 2: Design evaluations

The designs were evaluated using a modified card-sorting activity [22] in which participants viewed an empty main page and were presented with a list of the potential features and/or chapters they could include (Fig 2). These features and chapters were selected based on the interview theme analysis. The activity was conducted using Miro and participants were instructed to move the features and content to locations on the home page where they

**Table 1. List of Phase 1 semi-structured interview topics.**

| Topics | Topic Justification |
|---|---|
| Management of a TB case using the *Georgia TB Reference Guide* | A scenario where the *Guide* is needed informs the design, speed for information retrieval, and users' expectations. |
| Reference guide use settings | The type of tool users desire is dependent on where the information is accessed currently (I.e. in the office or on the hospital floor). |
| Guide navigation | The path for information retrieval in the book influences the navigation style in the digital version of the guide. |
| Guide customization (folded corners, notes, highlights) | How users modify the paper guide informs customization features in the digital version. Additionally, guide modifications inform the pathway for information retrieval (I.e. what users mark in the guide for memory is likely what they access often). |
| New edition management | The actions that users take when they receive a new edition of the guide defines how information should be structured to maintain continuity across content updates. |
| PDF version utilization | Utilization of the electronic version of the guide highlights users' expectations and familiarity with navigation patterns in electronic systems. Additionally, it demonstrates their thought process in deciding to use paper material vs electronic. |
| Supplementary reference tools | Knowing users' familiarity with medical reference tools helps inform what design patterns to mimic. Additionally, it helps identify how information is found in an app format. |

expected to find them. In addition, participants were told they could add new features, remove features, and add additional features or content sections. The visual structure of the app remained fixed as a guide for participants. Four individuals (3 TB coordinators at public health clinics and 1 ID Fellow) were recruited to participate in the card-sorting activity.

## Phase 3: App software development

The app was developed for the Android operating system using the Kotlin programming language and Android Studio (Google Inc, Palo Alto CA) integrated development environment (IDE), and for the iOS operating system using the Swift programming language and XCode (Apple Inc, Cupertino CA) IDE. While the application code is separated between the two operating systems, content pages are shared between the platforms using HyperText Markup Language (HTML) and Cascading Style Sheets (CSS). The PDF version of the guide was initially converted into HTML using Adobe Acrobat Pro (Adobe Inc, San Jose CA) and then refined for style and content. Furthermore, creating HTML content allowed for flexibility in creating a web version of the app in the future.

## Phase 3: Beta testing

We built and tested the app in two rounds to validate our translation of research insights into design implications and features. The two rounds were used to segment the development work into two smaller well-defined phases with the goal of collecting feedback before completing more of the app's development.

The first round focused on critical features: navigation, content structure, and search functionality; customization features were excluded. A cohort (n = 5, 3 ID fellows, 1 DPH nurse, and 1 attending physician) was recruited via email to participate in the beta test, which lasted between 20 to 30 minutes via Zoom. Participants were instructed to download the application on their personal devices prior to the call and to share their screen. Time on task was not

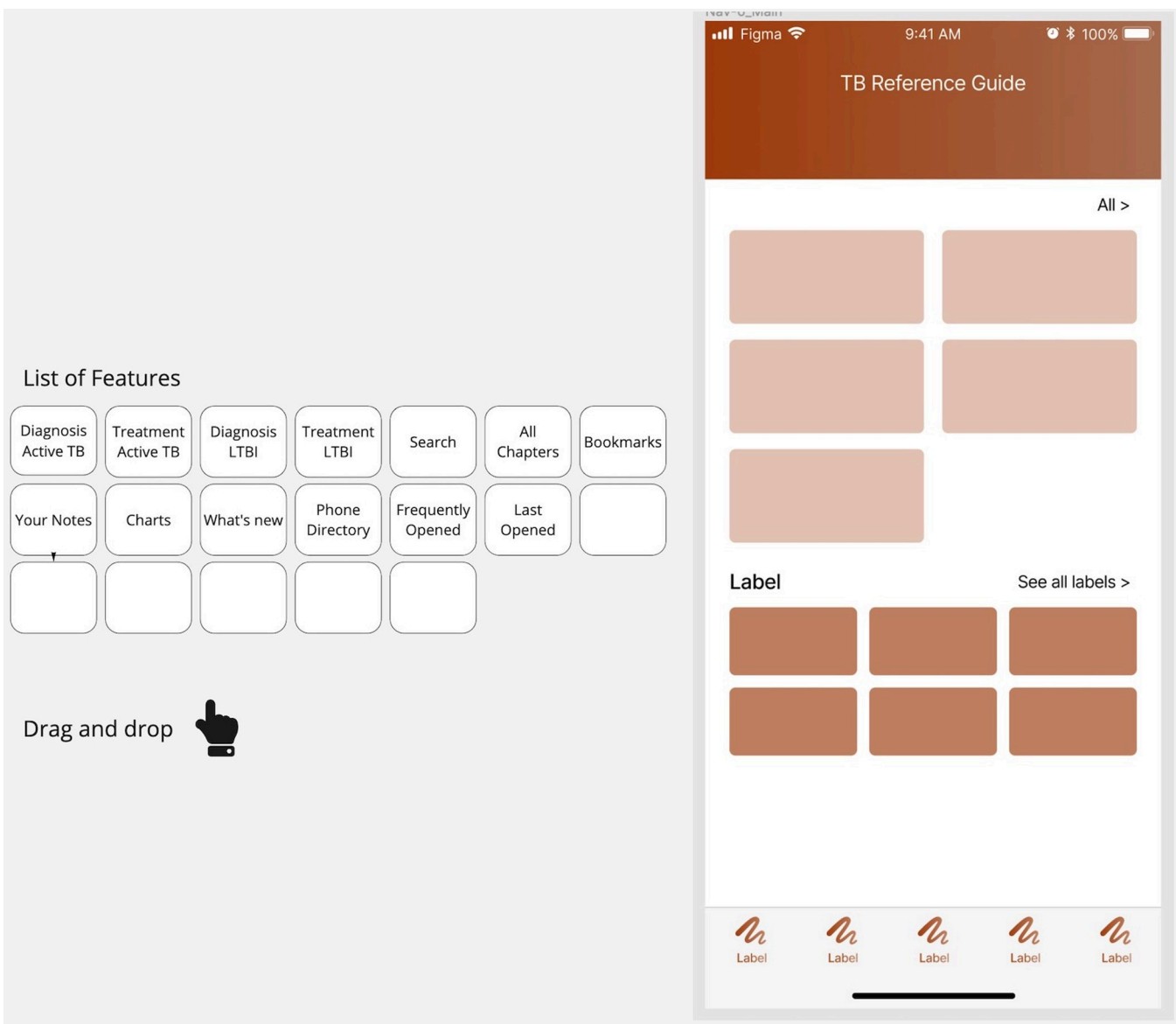

**Fig 2. Screenshot of the modified card-sorting activity.** Includes the list of features/content available and displays the home page layout without labels. Conducted using Miro (Miro Inc, San Francisco, CA).

collected. Qualitative questions were asked as part of both tests and occurred after the task to allow participants time to form opinions and judgements of the application.

The tests measured ease of access to information and identified areas of confusion. Participants completed a task-based evaluation which included the following tasks:

1. Looking for a piece of familiar content in the app

2. Looking for the chapter "Management of Children with Tuberculosis"

3. Looking for the HIV Treatment chapter (using search if it had not been used already)

Results from the first round of testing suggested changes to the main screen to improve usability as well as improving in-chapter references to other related content in the app.

The second round of beta testing assessed the changes implemented after the first round and tested the addition of the customization features, namely, bookmarking and adding notes to subchapters. A similar task-based evaluation from the previous round was used. An additional cohort (n = 5, 3 TB coordinators and 2 ID fellows) was recruited to participate in this round; two of the ID fellows participated in both rounds. Participants who already had the app installed were asked to update it between rounds.

Outcomes of the two rounds of beta testing suggested the app's features had been successfully designed to meet user expectations of how a digital version of the *Georgia TB Reference Guide* should work. This indicated that the app was ready for public release and an unmoderated round of testing.

### Phase 4: Survey

A cohort of individuals (n = 30) including the interviewed participants, other TB coordinators, and ID fellows were sent an email with a link to the official app available for download and a request to use the app and identify issues, crashes, or bugs. The survey asked the users to rate the app alone and compared to the book using a Likert Scale (1–5), and to provide any positive or negative feedback they had in the form of free-text entry. Although email recipients did not confirm their participation in using the app, the email invited recipients to share their thoughts in a future survey emailed 30 days later. A raffle of three $50 Amazon gift cards among survey respondents was used as an incentive. Qualtrics was used to create both the feedback survey as well as analyze the results.

### Phase 5a: Onboarding survey

A group of anonymous users (n = 61) completed an onboarding survey that appeared the first time the user opened the app. The survey asked users about their healthcare role, their primary practice setting, and the zip code of their primary practice. Answering these questions was optional and not a requisite to using the app.

### Phase 5b: Analytics

From February 1 to July 31, 2022, anonymous mobile app user analytics were collected (n = 281). The Google Firebase platform (Alphabet inc, Mountain View, CA) and the Pendo Analytics platform (Pendo.io, Inc. Raleigh, NC) were used for automated data collection and analysis. Data sharing is optional and adjustable from the users' personal devices and not a requisite to using the app.

## Results

### Phase 1 results: Interview findings

Interview data and insights revealed themes that validated our hypothesis for the need for a mobile app: 1) information gathering, 2) content availability, and 3) content usage.

**Validation: Users need a guide that is more frequently updated, is usable, and includes the same content as the book.**

1. Information gathering: Rate of information change is increasing

Interviewees with more than six years of experience in TB agreed that the rate of changes to state guidelines in TB was increasing in recent years. Due to this acceleration, new editions of the *Guide* became outdated more quickly and required users to manually track what information in the guide was no longer current.

2. Content availability: PDF version of the *Guide* is generally not utilized

Seven of the nine interviewees mentioned that they had seen the PDF version but did not use it and were not familiar with it. While an objective advantage of a PDF is the search feature which allows for quick information retrieval, no interviewees validated its utility within the PDF version. In addition, interviews identified one main drawback to the PDF version: the PDF was designed for use on a desktop computer, which limits its use to within an office setting; whereas the current guide is commonly used in clinical settings outside of the office.

3. Content Usage: Users provided positive feedback on the paper-copy *Guide*

Georgia has a go-to guide for TB that has been used and is loved by many of its users. Appreciation of the *Guide* grew with additional experience and usage; its most seasoned users (more than five years in the field) colloquially referred to it as the "TB Bible". All interviewees reported finding the guide easy to use, the information clear, and easily accessible.

*"This book knows what to do in different situations"* [P6]

*"The current guide has the most critical information in a concise manner, no fluff"* [P7]

*"This is easier than pulling up the nurse protocols, easy to share, train, and it fits in my pocket!"* [P3]

*"I like the font, the size, and weight of the book, med students are open to receiving information in this size"* [P5]

**Design: A more usable guide should facilitate access to common topics, be quick to navigate, and have additional space for users to add information.**

1. Treatment regimens and tables are frequently referenced

Content topics referencing numeric data and formulas were more frequently accessed than those primarily composed of narrative descriptions. This trend can be attributed to the inherently more challenging nature of recalling tabular data and calculations (for instance, recalling specific weight-based medication dosages) compared to remembering descriptive concepts (such as understanding what an AFB smear is).

2. Information is found by browsing, common use, or page markers

There was an even split between guide users who added markings to the guide—such as sticky notes in relevant pages or folded the corners of specific pages—and users who preferred keeping their guide unmarked. As an anecdote, one participant looked up the same content so frequently that her guide naturally opened to specific pages. Figures and headers were used for page navigation as well.

*"Charts and graphs tell you where you are"* [P5]

*"I use color flags, one color for LTBI, other for pediatrics, cases is blue, I mainly use the flags for awareness"* [P3]

Markings became an issue when a new guide of the version was distributed. Users described difficulty with transitioning to a new guide without their navigation markings. Six people would compare each regularly referenced section between the two guides to identify updates, and two participants reported using both new and old versions of the guide and referencing each edition depending on the clinical context.

*"Some people won't change to the new version because the old one had a lot of their notes"* [P5]

3. Blank space is needed to capture additional personal and important TB information
Interviews identified additional information streams such as email threads where updated information is stored.

*"the information lives in the bible [TB Guide], the state guidelines, the county protocols, and emails. The provider dependent guidelines stay in emails. There is a lot of information that simply lives in email threads"* [P5]

Some participants used separate notebooks or wrote in the *TB Guide* itself to supplement the information in the guide with things that are local and relevant to them.

*"I keep track of conversations [with providers, experts, patients] in a separate book. Usually, I first open the guide and then the diary [the separate book previously referenced]"* [P3]

*"in back of booklet [TB Guide] I keep contact info [the direct phone lines] for hospitals, jails, colleges, etc."* [P2]

## Phase 2 results: App features

In this section we describe how the design themes that emerged from interviews informed feature development within the mobile app.

**1. Static information displays benefit users with low use frequency.** In interviews, we asked participants how frequently they referred to the guide, their most-used content, marking preferences, and content importance. These findings directly informed a maximum-utility static information display. Each finding directly informed a design attribute, as follows.

Research Finding: Guide use is occasional, approximately four times per month or less.

- Design implication: Occasional use benefits from information layouts that are static and do not change, as opposed to dynamic layouts that change over time (such as having the most recently searched content appearing on the landing screen). Ideal formatting for occasional users would be to place commonly accessed topics prominently on the home page without needing additional clicks to access them.

- How design implication was implemented: Home page content was pre-defined based on most frequently used chapters and charts, and content is static. The most frequently used chapters and charts were determined using a card-sorting activity and included Diagnosis and Treatment of Active and Latent TB, and dosage charts.

**2. Search in the main page should emulate use patterns in common medical reference tools.** Participants were asked about their search behavior using other systems, how they searched using the PDF version of the guide, and their general approach to finding specific information quickly in the guide and other digital systems.

Research Finding: The card-sorting activity revealed that users wanted the ability to search within the main page as a secondary way of accessing the content.

- Design implication: Search should not be visually prioritized.

- How design implication was implemented: The search field was added to the top of the home page and a translucent background was used to decrease visual presence.

**3. Digital bookmarks, notes, and recent history were designed to mimic markings in the physical guide.**   As discussed above, participants were asked if and how they marked their paper guides, such as using highlighters, sticky notes, writing in margins, or dog-earing pages. Interviews also solicited what information was marked, how markings were used, and how markings shifted between editions of the guides.

Research Finding: As discussed in greater detail above, GDPH officials tended to add markings and notes to their guide to highlight specific pieces of information. Some of their markings included folding the pages, adding sticky notes, and writing in the guide or in a separate notebook.

- Design implication: App should allow customization via user markings and space to include additional content.

- How design implication was implemented: Users can bookmark a chapter or chart in the app similarly to marking by folding pages in a book. Additionally, users can add a note to a chapter or chart and color code the note to assign it additional meaning. These notes and bookmarks are accessible as a list within the app. Lastly, the app includes a history of most recently viewed chapters.

## Phase 3 results: Beta testing findings

The first round of app beta testing included the app's critical features: navigation, content, and search functionality. Once the app was available for download, five individuals were identified by the authors and emailed a link to download the app on their personal devices, and later participated in a video interview. The tests revealed needed improvements in search, main page, navigation, and content layout as outlined in Table 2.

The feedback was implemented in a period of two weeks and a new version of the guide was made available for download (Fig 3). Three new individuals were identified by the authors and followed the same procedure as in the first round of testing. Two individuals from the last round also participated. Existing users received a notification to update the app directly on their phones. The total number of individuals in the second round was also five.

The second round of beta testing demonstrated an effective improvement in information diversity on the main page, search functionality, and hyperlink navigation. The issues that were identified on the first round were not brought up again which validated that information diversity in the main page, search functionality, and hyperlinking navigation were improved. Additionally, all participants intuitively knew how to use the bookmarking and note saving features including editing and deleting created content.

## Phase 4 results: Survey data

From the cohort of individuals (n = 30) that received a link to complete a survey about their thoughts on the app, 60% (n = 18) completed the survey. Results showed that over 90% of participants rated the app useful both alone and compared to the physical book on the two Likert scale questions. Given the content is the same in both app and physical book, we attribute the app's high rating in usefulness to a successful migration of the content from a paper to a digital format.

The survey also invited free-text feedback on what app users liked and did not like about the app. All individuals provided positive feedback alluding to the app being convenient and easy to use, easy to navigate, and quickly accessible. More than half of the participants (n = 10) reported having no negative feedback. Two participants found the search feature within the app to be less effective than in the PDF version. Another participant wanted the app to be less

**Table 2. Design alterations based on findings from first round of evaluations.**

| Finding | Design Alterations | Quotes |
|---|---|---|
| **Search functionality** - Search needs to speed up information retrieval | Search functionality can use either one keyword i.e. "pediatric" or multiple keywords to increase specificity i.e. "active pediatric HIV" <br> Search results should prioritize displaying the chapter and subchapter titles over the summary text | No specific quotes, rather participant behavior revealed they commonly typed multiple terms in one search query to look for more specific content |
| **Content** - Main page utility depends on content diversity | Charts displayed in the main page should be diverse rather than reference similar topics | *"I like special charts like extra-pulmonary, [but] right now three charts are about first-line TB treatment"* P4 |
| **Content** - Main page needs to be clear on amount of app content | Labels in the main page directing towards all chapters and charts need to be more prominent | *"I didn't see the 'all chapters' label"* P1, P4, P5 <br> *"Didn't know charts is just the tables for the chapters"* P1, P2 <br> *"There's no indication that these are select chapters"* P1, P2 |
| **Navigation** - Navigation in the app needs to be improved to find referenced content | Content pages should include more references to each other to allow within-chapter navigation, i.e. related topics links <br> Introduction subchapter should include links to tables and figures | *"UTD [UpToDate] articles contain summary with subchapter links but also figures and chart links"* P5 |
| **Content** - Content needs to be adjusted to facilitate reading | Content pages should include subheaders to facilitate reading <br> Charts should auto-zoom to display 100% of the size | *"Content that expands for pages I won't read"* [related to pieces of content that span multiple pages in the book] P3 <br> *"Is it possible to have subheaders for content i.e. what is the TST? How is it performed? As a way to identify pockets of interest"* P5 <br> *"Don't want to scroll on tables sideways to view content"* P1, P3 |

text-heavy, and two participants found the tables difficult to read due to organization or small font size. These comments were shared with the *TB Guide*'s editorial team for further consideration.

## Phase 5 results: In-app survey data and usage analytics

The number of users of the *Georgia TB Reference Guide* app grew from 30 pre-selected individuals to 281 in the United States at the six-month mark following its launch on January 31, 2022. Of these, there was a relatively even split between iOS and Android. While most of the downloads (125) were in Georgia as would be expected, the app was also downloaded in 19 other states including California (56), Texas (12), and Kansas (11).

During the initial six months, the average user was active on the app for 2.5 days and used the app for 9.4 minutes, while the median user was active 1 day and used the app for 5.0 minutes. The most popular content page was the treatment of active disease with 1,394 page views accessed by 124 unique users. A detailed description of the app's usage will be the subject of a future publication.

The image above (Fig 4) shows dissemination across the United States in the first six months post-app release. During the first month after the app's release, there were downloads in six other states: Texas, California, Kansas, Florida, Alabama, Tennessee, and New York. After the first six months, there were app downloads in 19 different states. This is notable because advertisement for the app was limited within the first six months. All growth occurred

**Box 1. Additional findings and changes**

1. An optical illusion in the home page that was distracting to users — black dots would appear in between the chapter cards, this phenomenon is called the Scintillating Grid Illusion [2]. We incorporated a visual metaphor, "dog-ears" or folds on the pages of a book, to the chapter and subchapter cards, as well as shadowing.

2. Some participants missed the "All Chapters" button on the top right but they did see the "See all Charts" down below. A "Chapters" label was added to the left side to draw attention to that header visually.

3. The search field was not very visible due to its translucent background. It was changed to pure white to increase visual relevancy.

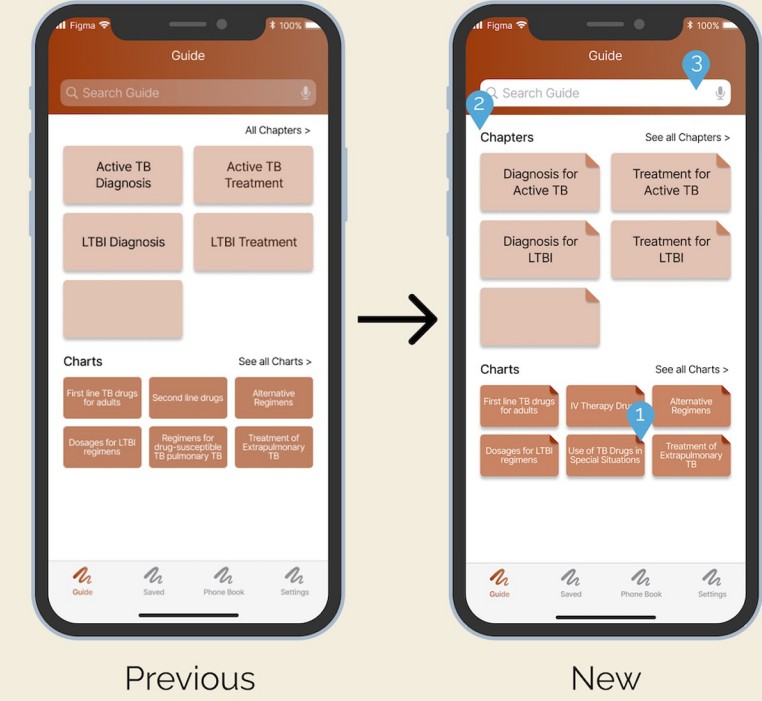

**Fig 3. Screenshots of the previous design and new design.** Includes annotations of change. Designs done in Figma (Figma Inc, San Francisco CA).

through word of mouth, suggesting high utility and demand for mobile-friendly digital TB reference tools across the US.

At the six-month mark post-release, 61 anonymous users (21%) completed the voluntary onboarding survey within the app (Fig 4, bottom-right). Most respondents were nurses (54%) and physicians (25%) working in public health departments (62% of the 48 nurses/physicians) and academic center inpatient settings (22%).

The content of the app was updated twice in the six months following its release on the app stores. Relevant pieces of updated content included:

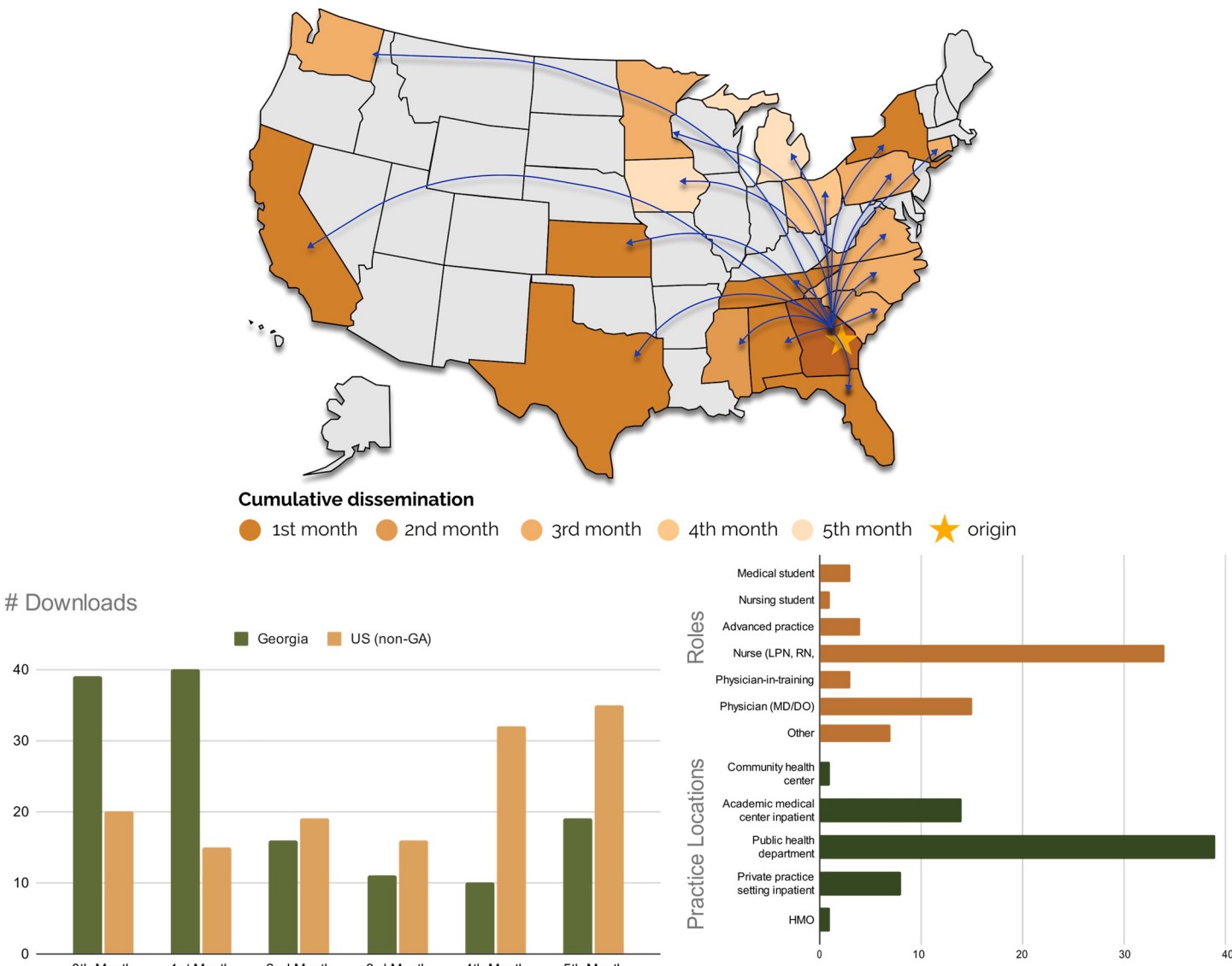

**Fig 4. Usage analytics.** Top-left: month by month cumulative visualization of the app dissemination across the US. Bottom-left: Download numbers from Georgia and US (excepting Georgia) on a month-to-month basis. Bottom-right: Sample of users' demographics on healthcare role and location of practice at the six-month mark post release (n = 61).

1. Sharing a new study on active TB treatment that could shorten the treatment length from six months to four months

2. Updating the reference material

3. Updating the contact information for one Georgia state TB coordinator

4. Updating epidemiology statistics from 2018 to 2020

While the paper-based guidebook was only updated every few years, the app made two updates possible within the first six months, underscoring the immensely improved ease in updating a mobile app guidebook compared to a paper-based guidebook. In addition to its speed, releasing updates to the mobile app guidebook is significantly more cost effective.

Lastly, automatic updates address a previous user pain point of needing to transfer individual markings from one version to another. After updates, all existing markings remain unaffected.

## Discussion

Our work in transforming the *Georgia TB Reference Guide* from a traditional paper format into a mobile application represents a significant advancement in the dissemination of medical information. The successful uptake of the app, as evidenced by its adoption across multiple states, underscores the growing need for accessible, user-friendly digital resources in the healthcare field. This transition from paper to digital not only enhances the ease of access and use but also ensures timely updates of critical medical information, which is paramount in a rapidly evolving field like TB treatment.

The detailed process we presented serves as a blueprint for similar endeavors in other medical disciplines. Commencing the project interviews with current guide users helped gain a better understanding of their utilization practices. This information then informed the design of the app in the second phase, where preliminary app designs were created and evaluated through further user interviews. Two rounds of beta testing with early versions of the app allowed for early identification of bugs and issues and validating the designs did meet user expectations. This helped further refine the app's features and functionality to ensure its intuitive use and adoption. Once the app was publicly launched in the application stores, surveying paper-based guide users helped gather feedback on their experiences using the app and how it compared with the guide. This feedback was used to further validate that the conversion from paper to digital was effective. Each phase of the app's development was key, providing critical feedback at each stage, and validating each decision and assumption along the way.

The organic dissemination of the app beyond the state of Georgia's borders within the first six months of launch underscores the desire among nationwide-TB healthcare professionals for high-quality digital reference content. The app's reach extended to 19 states, including Texas, California, and Kansas, with a total of 281 downloads in the United States. Interestingly, most downloads outside Georgia occurred through organic dissemination, indicating that mobile apps can serve as an effective tool for disseminating and promoting awareness of TB reference guidelines.

However, the process and solution described does have some challenges and limitations. Dependence on internet connectivity and battery constraints should also be considered when designing online reference material mobile systems, particularly in remote or resource-limited settings. In our case, since we were focused on urban as well as rural dissemination, we addressed the connectivity issue by storing all content locally on the user's phone and only needing internet to update to new versions of the app. Addition challenges include the dynamic nature of mobile operating systems, like iOS and Android, necessitating regular updates to the app. This not only involves technical challenges but also incurs ongoing costs and resource allocation to ensure compatibility and smooth functionality with each new software version.

Since a reference tool for TB is typically consulted only occasionally, closer to monthly than daily, this infrequent usage might result in a slower identification and resolution of technical problems compared to applications that are used more regularly. This could potentially impact the usability and effectiveness of the app. Additionally, the survey responses collected in phases 4 and 5 may have been subject to respondent bias, as only high-frequency users of the app may have participated in the surveys. Also, the primary research phase was largely based on TB coordinator experiences with the guide, which differs from other user groups such as

physicians-in-training or inpatient nurses. These limitations should be considered when evaluating the overall success of the app and identifying areas for future improvement.

In conclusion, the transition of the *Georgia TB Reference Guide* into a mobile app not only represents a significant step in modernizing medical reference materials but also sets a precedent for future digital transformations in healthcare. As digital technologies continue to permeate the healthcare sector, mobile applications like ours will become increasingly vital in disseminating up-to-date medical knowledge, fostering best practices, and ultimately enhancing patient care.

## Conclusion

This paper highlights the successful development and implementation of a user-centered digital reference guide for TB management in the state of Georgia. The guide was created based on the needs and feedback of healthcare workers, validated through every phase, resulting in an intuitive and practical tool that has seen promising rates of adoption at the state and national levels. The organic expansion of the app to 19 states and more than 281 users underscores the demand for reliable and accessible resources for managing TB. This manuscript offers valuable insights into the importance of user-centered design in creating effective clinical tools and provides a blueprint for future digital reference guide development projects in healthcare.

## Acknowledgments

The authors would like to acknowledge the contribution of Comfort Mwalija in the development of the Android version of the *Georgia TB Reference Guide* mobile application. The authors would also like to acknowledge the work of Maren Parsell in assisting with data collection for the primary research. Lastly the authors would like to acknowledge the help of all individuals who were interviewed and provided invaluable information and feedback throughout the different stages of the project, it would not have been possible without their engagement and support.

## Author Contributions

**Conceptualization:** Santiago J. Arconada Alvarez, Morgan Greenleaf.

**Data curation:** Santiago J. Arconada Alvarez, Morgan Greenleaf.

**Formal analysis:** Santiago J. Arconada Alvarez, Alison T. Hoover.

**Funding acquisition:** Henry M. Blumberg.

**Investigation:** Santiago J. Arconada Alvarez.

**Methodology:** Santiago J. Arconada Alvarez, Alison T. Hoover.

**Project administration:** Morgan Greenleaf, Wilbur A. Lam.

**Software:** Santiago J. Arconada Alvarez.

**Supervision:** Morgan Greenleaf, Susan M. Ray, Marcos C. Schechter, Henry M. Blumberg.

**Validation:** Santiago J. Arconada Alvarez.

**Visualization:** Santiago J. Arconada Alvarez.

**Writing – original draft:** Santiago J. Arconada Alvarez, Alison T. Hoover, Morgan Greenleaf.

**Writing – review & editing:** Marcos C. Schechter, Henry M. Blumberg, Wilbur A. Lam.

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
