## [Decision Letter · Decision Letter 0]

17 Oct 2023

PONE-D-23-18533“An app is just available at all times” - the process and outcomes of converting the Georgia Tuberculosis Reference Guide into a Mobile ApplicationPLOS ONE

Dear Dr. Lam,

Thank you for submitting your manuscript to PLOS ONE. After careful consideration, we feel that it has merit but does not fully meet PLOS ONE’s publication criteria as it currently stands. Therefore, we invite you to submit a revised version of the manuscript that addresses the points raised during the review process.

We look forward to receiving your revised manuscript.

Kind regards,

Anna Bernasconi, PhD

Academic Editor

PLOS ONE

Journal Requirements:

2. Thank you for including your ethics statement:  "N/A".   

A) For studies reporting research involving human participants, PLOS ONE requires authors to confirm that this specific study was reviewed and approved by an institutional review board (ethics committee) before the study began. Please provide the specific name of the ethics committee/IRB that approved your study, or explain why you did not seek approval in this case.

B) Please provide additional details regarding participant consent. In the ethics statement in the Methods and online submission information, please ensure that you have specified (1) whether consent was informed and (2) what type you obtained (for instance, written or verbal, and if verbal, how it was documented and witnessed). If your study included minors, state whether you obtained consent from parents or guardians. If the need for consent was waived by the ethics committee, please include this information.

**Additional Editor Comments:**

Dear authors, the work was generally appreciated by the reviewers. Several comments for improvements are made. Please revise your manuscript accordingly and re-submit it highlighting the changed/added parts.

Reviewers' comments:

Reviewer's Responses to Questions

**Comments to the Author**

1. Is the manuscript technically sound, and do the data support the conclusions?

Reviewer #1: Yes

Reviewer #2: Partly

Reviewer #3: Yes

Reviewer #4: Yes

2. Has the statistical analysis been performed appropriately and rigorously? 

Reviewer #1: Yes

Reviewer #2: Yes

Reviewer #3: N/A

Reviewer #4: N/A

3. Have the authors made all data underlying the findings in their manuscript fully available?

Reviewer #1: Yes

Reviewer #2: Yes

Reviewer #3: Yes

Reviewer #4: Yes

4. Is the manuscript presented in an intelligible fashion and written in standard English?

Reviewer #1: Yes

Reviewer #2: Yes

Reviewer #3: Yes

Reviewer #4: Yes

5. Review Comments to the Author

Reviewer #1: This is an interesting paper that continues to highlight the importance of mobile health app in the healthcare industry and more so the specific area of tuberculosis. The choice of the design method was particularly important ( user centered design using a mixed methods approach) as the end result showed a 90% preference of the app over the paper versions of the guide. The fact that there is potential wide adaptation of the app by more states would in my view have necessitated an IRB approval but since the authors have adequately provided a waiver for the IRB requirement I think it is adequate to allow wide adaptations

Reviewer #2: This work is interesting and has impacts on the health apps development. However, I have some comments, please see below.

1. In the introduction section, regarding “A survey of the available mobile apps related to tuberculosis found that although the number of available apps has increased, most of them contained errors, or provided wrong or harmful information [13]. This identified the existence of a large market gap for high-quality clinician-driven information.” This expression is quite vague as to what kind of wrong or harmful information it refers to. Please elaborate or provide some examples because this is an issue with the existing TB mobile apps. A clear explanation is needed.

2. Also in the introduction section, the last paragraph “To address this, we conducted a series of user-centered qualitative interviews and quantitative surveys to understand current users, design, develop, and validate the utility of a mobile application version of the Georgia TB Reference Guide mobile application.” I think this beginning is very abrupt, probably because you didn't make it clear what the issue at hand was in the previous paragraph (see comment 1). I know you want to emphasize "user-centered" which is good, but it came out of nowhere and I would suggest rewording this to make it logical. And also explain what “user-centered” means in your study.

3. “To our knowledge this is the first study delineating the process and value of transitioning a paper-based reference guide into a mobile application, particularly in the field of TB.” I feel this argument is too arbitrary, although I didn't go to verify but during COVID-19 there are some practices of putting guides into mobile apps. I would suggest removing this argument to avoid being too arbitrary.

4. This “Figure 1. Flow chart of the study’s methodology” looks very good, but upload a high-resolution one. And please note you have two figure 1.

5. In Phase 2: Design Evaluations section, I am curious why you evaluate the designs with a modified card-sorting activity, is there any reason behind this or is this method better than others?

6. The authors only asked the users to answer the onboarding survey about health care role, their primary practice setting, and the zip code of their primary practice. Why not ask for any perceptions of using this app? Because the questions you are asking are more about the user’s portrait, why knowing this is important rather than evaluating this app?

7. Regarding the results section, I noticed that most of the findings were qualitative, but I didn't see any description of how they were analyzed, how did you generate these findings? Inductive analysis, some coding work or something else?

8. I think you should emphasize why your work (converting paper guide into mobile app) is important and what's the takeaway in the discussion section.

Reviewer #3: The submitted manuscript provides readers insight to the development of a mobile app based on a paper clinical reference guide for TB care. The outlined process is systematic and detailed in such a way to provide a guide for others.

There are a few minor typos/duplicative words for the authors to address.

• In the abstract’s methods section the authors outline 5 step process, but label 2 steps as “fourth”. Please change the first word of the last sentence in this section to “Fifth.”

• In the first sentence of the last paragraph of the introduction: “To address this, we conducted a series of user-centered qualitative interviews and quantitative surveys to understand current users, design, develop, and validate the utility of a mobile application version of the Georgia TB Reference Guide mobile application.”, suggest removing the words “mobile application” at the end of the sentence, as it is duplicative of information provided earlier in the sentence.

• Under methods, Phase 3: Beta Testing the following sentence needs the word “to” added between the words “recruited participated” – “An additional cohort (n =5, 3 TB coordinators and 2 ID fellows) was recruited participated in this round; two of the ID fellows participated in both rounds.”

For clarification, the authors are encouraged to consider the following.

• Add a footnote with references for Figma, Swift and Kotlin, and Pendo below Figure 1 – Flow chart of the study’s methodology.

• Methods Section - “Phase 1: Interviews” define or provide an example of “low-detail design,” noted in the statement “Themes were translated into design implications and low-detail designs were produced using the digital design tool Figma (Figma Inc, San Francisco CA)”

• Methods Section - “Phase 2: Design Evaluations” – The following statement is grammatically incorrect. “A group was individuals (n= 4, 3 DPH coordinators & 1 ID fellow) was recruited to participate in this card sorting activity” Further, a “group” indicates several persons, four persons is a rather small group. Suggest re-writing this statement as “Four individuals (3 DPH coordinators, 1 ID Fellow) were recruited to participate in this card sorting activity.

• Results section - “Phase 1 Results: Interview Findings” the authors list 4 themes: validation, information gathering, content usage, and content availability.” The following sections are titled “Validation” and “Design” …. For clarity, label the sections with the themes noted under the design section.

• Results section - define or provide an example of “content topics referencing numbers.”

• Results section - Table 2, add a column and specify which of the beta testing finding themes “navigation” “content” “search functionality” apply to the rows in the table.

• Results section - Table 2, expand the following quote to provide a reader more context - “Content that expands for pages I won’t read.”

• The authors write, “The second round of beta testing proved effective at improving information diversity in the main page, search functionality, and hyperlinking navigation. The issues that were identified on the first round were not brought up again which validates that information diversity in the main page, search functionality, and hyperlinking navigation had all been improved.” The first sentence needs to be edited as the second round of testing did not accomplish an effect, but rather demonstrated an effect following the first round of testing.

• The section “Phase 4: Survey” provides little information about the survey itself. How many questions were asked, what was the format of the questions? Correspondingly, section “Phase 4 survey results: Survey Data” provides little clarifying information. First, add the (n=18) alongside 60% in the first sentence of this section. The second sentence notes app user’s response to 1 or 2 questions regarding usefulness of the app alone, and compared to the hardcopy book. Are these the only close-ended questions asked? If so, please specify this. The 2nd paragraph notes responses to free text feedback. These types of questions typically have low response rates. How many of the 18 respondents provided free-text feedback, and how many respondents are reflected in the summary statements.

• Phase 5 Results: In the first sentence of the first paragraph, suggest adding the words “number of users” prior to the words “Georgia TB Reference Guide.”

• Phase 5 Results: the second paragraph clarify “During that period” …. Is this “During the first X months the app was available.”

• The drawbacks of a paper guide are outlined in this manuscript. The Discussion section would be strengthened by adding information about any challenges that may arise when a user has limited connectivity to wi-fi or low battery reserves on their phone, as well challenges and costs associated with keeping the app updated following iOS updates, etc. To add this information, the authors could edit information in the Conclusion Section that is duplicative of information provided in the introduction and results section.

Reviewer #4: This is a useful article detailing the process of converting medical guidance to an electronic format. The example of the GDPH TB guidance is illustrative and helpful. There are a number of minor copy-edits to make and recommend not referring to the process as a study.

Minor edits:

- page 13: Phase 2: Design Evaluations paragraph:...A group OF individuals (n=4, 3DPH coordinators...)

- page 14: Phase 3: Beta Testing: "... to improve usability as well as improving...

- page 14: Phase 3: Beta Testing: "An additional cohort was recruited AND participated..."

- page 14: Phase 4: Survey: "...did not confirm their participation using the app..."

- page 15: 3. Users provided positive feedback on the current TB guide: "... that has been used and is loved by man its users."

- page 15: 3. Users provided positive feedback on the current TB guide: "... its most seasoned users..."

- page 16: 2. Information is found by browsing, common use, or page markers: recommend providing actual numbers instead of using "few" and "couple"

- page 17 (first line): recommend rephrasing "... Some participants used separate notebooks or wrote TO the TB Guide itself..."

- page 20: Box 1, 1st recommendation: the other apparent addition was the shadowing of the boxes

- page 23: Discussion, first paragraph: recommend not referring to this as study

6. PLOS authors have the option to publish the peer review history of their article (what does this mean?). If published, this will include your full peer review and any attached files.

Reviewer #1: **Yes: **James Akiruga Amisi

Reviewer #2: **Yes: **Huanyu Bao

Reviewer #3: No

Reviewer #4: No

---

## [Author Response · Author response to Decision Letter 0]

16 Jan 2024

E1 RESPONSE: 

We have reviewed and ensured that our manuscript adheres to PLOS ONE's style requirements. 

2. Thank you for including your ethics statement: "N/A". 

A) For studies reporting research involving human participants, PLOS ONE requires authors to confirm that this specific study was reviewed and approved by an institutional review board (ethics committee) before the study began. Please provide the specific name of the ethics committee/IRB that approved your study, or explain why you did not seek approval in this case. 

B) Please provide additional details regarding participant consent. In the ethics statement in the Methods and online submission information, please ensure that you have specified (1) whether consent was informed and (2) what type you obtained (for instance, written or verbal, and if verbal, how it was documented and witnessed). If your study included minors, state whether you obtained consent from parents or guardians. If the need for consent was waived by the ethics committee, please include this information. 

E2 RESPONSE: 

All participants consented verbally to participating in the research. The project was reviewed by the Emory University Institutional Review Board and determined to not require IRB review because it did not meet the definition of “research” with human subjects, or “clinical investigation” as set forth in Emory policies and procedures and federal rules. This information has been added to the Methods section and the online submission information. 

E3 RESPONSE: 

All data is available under the following DOI 

https://doi.org/10.5281/zenodo.10253131

E4 Response: 

We have thoroughly reviewed our reference list to ensure its completeness and accuracy. We have also two more references [x1] and [x2] to support additional evidence on the ‘user centered design’ and ‘inductive reasoning’ as suggested by the reviewers. 

Additional Editor Comments: 

Dear authors, the work was generally appreciated by the reviewers. Several comments for improvements are made. Please revise your manuscript accordingly and re-submit it highlighting the changed/added parts. 

Reviewers' comments: 

Reviewer's Responses to Questions 

Comments to the Author 

1. Is the manuscript technically sound, and do the data support the conclusions? 

Reviewer #1: Yes 

Reviewer #2: Partly 

Reviewer #3: Yes 

Reviewer #4: Yes 

2. Has the statistical analysis been performed appropriately and rigorously? 

Reviewer #1: Yes 

Reviewer #2: Yes 

Reviewer #3: N/A 

Reviewer #4: N/A 

3. Have the authors made all data underlying the findings in their manuscript fully available? 

Reviewer #1: Yes 

Reviewer #2: Yes 

Reviewer #3: Yes 

Reviewer #4: Yes 

4. Is the manuscript presented in an intelligible fashion and written in standard English? 

Reviewer #1: Yes 

Reviewer #2: Yes 

Reviewer #3: Yes 

Reviewer #4: Yes 

5. Review Comments to the Author 

Reviewer #1: This is an interesting paper that continues to highlight the importance of mobile health app in the healthcare industry and more so the specific area of tuberculosis. The choice of the design method was particularly important ( user centered design using a mixed methods approach) as the end result showed a 90% preference of the app over the paper versions of the guide. The fact that there is potential wide adaptation of the app by more states would in my view have necessitated an IRB approval but since the authors have adequately provided a waiver for the IRB requirement I think it is adequate to allow wide adaptations 

R1-RESPONSE: 

Thank you for your review. 

Reviewer #2: This work is interesting and has impacts on the health apps development. However, I have some comments, please see below. 

1. In the introduction section, regarding “A survey of the available mobile apps related to tuberculosis found that although the number of available apps has increased, most of them contained errors, or provided wrong or harmful information [13]. This identified the existence of a large market gap for high-quality clinician-driven information.” This expression is quite vague as to what kind of wrong or harmful information it refers to. Please elaborate or provide some examples because this is an issue with the existing TB mobile apps. A clear explanation is needed. 

R2-1 RESPONSE: 

We have added the following to the manuscript: This review [13] identified incorrect descriptions of tuberculosis describing it as diabetes mellitus, erroneous medical advice such as alluding to apple custard remedies lacking up-to-date information, and spelling or grammatical mistakes. 

2. Also in the introduction section, the last paragraph “To address this, we conducted a series of user-centered qualitative interviews and quantitative surveys to understand current users, design, develop, and validate the utility of a mobile application version of the Georgia TB Reference Guide mobile application.” I think this beginning is very abrupt, probably because you didn't make it clear what the issue at hand was in the previous paragraph (see comment 1). I know you want to emphasize "user-centered" which is good, but it came out of nowhere and I would suggest rewording this to make it logical. And also explain what “user-centered” means in your study. 

R2-2 RESPONSE: 

The introduction has been revised to include a clearer transition into the discussion of our user-centered approach, along with an explanation of what 'user-centered' means in the context of our study. We added the following text. “To address this gap, we focused on developing a user-friendly mobile application version of the Georgia TB Reference Guide, to leverage its clinical validity and make it into an accessible format. A 'user-centered' approach underpinned our methodology, emphasizing the importance of understanding and incorporating the needs, preferences, and experiences of the end-users - in this case, healthcare professionals dealing with TB. This approach is characterized by active and continuous engagement with users through qualitative interviews and quantitative surveys. This engagement is directed at understanding current users, designing, developing, and validating the utility of, in our case, a mobile application [x1].” 

3. “To our knowledge this is the first study delineating the process and value of transitioning a paper-based reference guide into a mobile application, particularly in the field of TB.” I feel this argument is too arbitrary, although I didn't go to verify but during COVID-19 there are some practices of putting guides into mobile apps. I would suggest removing this argument to avoid being too arbitrary. 

R2-3 RESPONSE: 

We have removed that argument from the manuscript. 

4. This “Figure 1. Flow chart of the study’s methodology” looks very good, but upload a high-resolution one. And please note you have two figure 1. 

R2-4 RESPONSE: 

We have uploaded a higher resolution version of the figure. 

5. In Phase 2: Design Evaluations section, I am curious why you evaluate the designs with a modified card-sorting activity, is there any reason behind this or is this method better than others? 

R2-5 RESPONSE: 

Definitely. We chose the method because it was restrictive in nature by providing a template of the home page and only allowing the participants to visually rank the features and topic areas of the guide. This helped us determine what content they found more relevant while keeping it restricted to a small visual space. Additionally, it allowed us to validate the findings from the user research phase and determine whether there were any particular features that users didn’t feel the need for. As an anecdote, our research phase showed that knowing what content had changed was important, but none of the participants placed that tile in any areas of importance. 

6. The authors only asked the users to answer the onboarding survey about health care role, their primary practice setting, and the zip code of their primary practice. Why not ask for any perceptions of using this app? Because the questions you are asking are more about the user’s portrait, why knowing this is important rather than evaluating this app? 

R2-6 RESPONSE: 

Good question and very valid. For this phase of the project and the first release of the app we wanted to focus on adoption and avoid having the user answer unnecessary questions to get to the content. An app asking you for your healthcare role feels relevant to your purpose of using the app (i.e. getting to the content), however the app asking whether you like it or not meets the purpose only for the researcher. Additionally keep in mind this survey was launched upon *opening* the app. It would not have been appropriate to assess the utility of the app during the first or second use. It is a survey that can and should be launched at a later stage, after sufficient use of the app. In this instance, we started with the pertinent questions during early phase use of the app, which will be followed up by perceived utility surveys. 

7. Regarding the results section, I noticed that most of the findings were qualitative, but I didn't see any description of how they were analyzed, how did you generate these findings? Inductive analysis, some coding work or something else? 

R2-7 RESPONSE: 

We used an inductive analysis process paired with the affinity diagramming technique to organize common themes together using Miro. Have included it in the manuscript to specific and incorporated citation [x2] 

8. I think you should emphasize why your work (converting paper guide into mobile app) is important and what's the takeaway in the discussion section. 

R2-8 RESPONSE: 

The discussion has been revised to more prominently highlight the significance and key takeaways of converting the paper guide into a mobile app. 

Reviewer #3: The submitted manuscript provides readers insight to the development of a mobile app based on a paper clinical reference guide for TB care. The outlined process is systematic and detailed in such a way to provide a guide for others. 

There are a few minor typos/duplicative words for the authors to address. 

• In the abstract’s methods section the authors outline 5 step process, but label 2 steps as “fourth”. Please change the first word of the last sentence in this section to “Fifth.” 

• In the first sentence of the last paragraph of the introduction: “To address this, we conducted a series of user-centered qualitative interviews and quantitative surveys to understand current users, design, develop, and validate the utility of a mobile application version of the Georgia TB Reference Guide mobile application.”, suggest removing the words “mobile application” at the end of the sentence, as it is duplicative of information provided earlier in the sentence. 

• Under methods, Phase 3: Beta Testing the following sentence needs the word “to” added between the words “recruited participated” – “An additional cohort (n =5, 3 TB coordinators and 2 ID fellows) was recruited participated in this round; two of the ID fellows participated in both rounds.” 

R3-1 RESPONSE: 

All identified typos and duplicative words have been corrected as suggested. 

For clarification, the authors are encouraged to consider the f

---

## [Decision Letter · Decision Letter 1]

31 Jan 2024

“An app is just available at all times” - the process and outcomes of converting the Georgia Tuberculosis Reference Guide into a Mobile Application

PONE-D-23-18533R1

Dear Dr. Lam,

We’re pleased to inform you that your manuscript has been judged scientifically suitable for publication and will be formally accepted for publication once it meets all outstanding technical requirements.

Kind regards,

Anna Bernasconi, PhD

Academic Editor

PLOS ONE

Additional Editor Comments (optional):

The authors have produced an extensively edited manuscript, which now responds to all the comments and concerns of the reviewers. The quality of the manuscript now reaches the standards of the journal and I recommend it is accepted.

Reviewers' comments:

Reviewer's Responses to Questions

**Comments to the Author**

1. If the authors have adequately addressed your comments raised in a previous round of review and you feel that this manuscript is now acceptable for publication, you may indicate that here to bypass the “Comments to the Author” section, enter your conflict of interest statement in the “Confidential to Editor” section, and submit your "Accept" recommendation.

Reviewer #2: All comments have been addressed

2. Is the manuscript technically sound, and do the data support the conclusions?

Reviewer #2: Yes

3. Has the statistical analysis been performed appropriately and rigorously? 

Reviewer #2: Yes

4. Have the authors made all data underlying the findings in their manuscript fully available?

Reviewer #2: Yes

5. Is the manuscript presented in an intelligible fashion and written in standard English?

Reviewer #2: Yes

6. Review Comments to the Author

Reviewer #2: I believe the author has effectively addressed the comments I made earlier. The revised manuscript looks much improved.

7. PLOS authors have the option to publish the peer review history of their article (what does this mean?). If published, this will include your full peer review and any attached files.

Reviewer #2: No
